# Can We Improve Mortality Prediction in Patients with Sepsis in the Emergency Department?

**DOI:** 10.3390/medicina60081333

**Published:** 2024-08-16

**Authors:** Sonia Luka, Adela Golea, Ștefan Cristian Vesa, Crina-Elena Leahu, Raluca Zăgănescu, Daniela Ionescu

**Affiliations:** 1Department 6 Surgery, Discipline of Emergency Medicine, Iuliu Hatieganu, Faculty of Medicine, University of Medicine and Pharmacy, 3–5 Clinicilor Street, 400347 Cluj-Napoca, Romania; adela.golea@umfcluj.ro; 2Clinical Emergency County Hospital, 3–5 Clinicilor Street, 400347 Cluj-Napoca, Romania; leahu.crina.elena@elearn.umfcluj.ro (C.-E.L.); zaganescu.raluca@elearn.umfcluj.ro (R.Z.); 3Department 1 Functional Sciences, Discipline of Pharmacology, Toxicology and Clinical Pharmacology, Faculty of Medicine, Iuliu Hatieganu University of Medicine and Pharmacy, 23 Marinescu Street, 400337 Cluj-Napoca, Romania; stefan.vesa@umfcluj.ro; 4Department 6 Surgery, Discipline of Anesthesia and Intensive Care I, Faculty of Medicine, Iuliu Hatieganu University of Medicine and Pharmacy, 19–21 Croitorilor Street, 400162 Cluj-Napoca, Romania; daniela_ionescu@umfcluj.ro; 5Department of Anesthesia and Intensive Care, The Regional Institute of Gastroenterology and Hepatology, “Prof. Dr. Octavian Fodor”, 19–21 Croitorilor Street, 400162 Cluj-Napoca, Romania; 6Research Association in Anesthesia and Intensive Care (ACATI), 400394 Cluj-Napoca, Romania; 7Outcome Research Consortium, Cleveland, OH 44195, USA

**Keywords:** biomarkers, emergency department, interleukin-6, prognosis, sepsis

## Abstract

*Background and Objectives*: Sepsis represents a global health challenge and requires advanced diagnostic and prognostic approaches due to its elevated rate of morbidity and fatality. Our study aimed to assess the value of a novel set of six biomarkers combined with severity scores in predicting 28 day mortality among patients presenting with sepsis in the Emergency Department (ED). *Materials and Methods*: This single-center, observational, prospective cohort included sixty-seven consecutive patients with septic shock and sepsis enrolled from November 2020 to December 2022, categorized into survival and non-survival groups based on outcomes. The following were assessed: procalcitonin (PCT), soluble Triggering Receptor Expressed on Myeloid Cells-1 (sTREM-1), the soluble form of the urokinase plasminogen activator receptor (suPAR), high-sensitivity C-reactive protein (hs-CRP), interleukin-6 (IL-6), and azurocidin 1 (AZU1), alongside clinical scores such as the Quick Sequential Organ Failure Assessment (qSOFA), Systemic Inflammatory Response Syndrome (SIRS), the Sequential Organ Failure Assessment (SOFA), the Acute Physiology and Chronic Health Evaluation II (APACHE II), the Simplified Acute Physiology Score II and III (SAPS II/III), the National Early Warning Score (NEWS), Mortality in Emergency Department Sepsis (MEDS), the Charlson Comorbidity Index (CCI), and the Glasgow Coma Scale (GCS). The ability of each biomarker and clinical score and their combinations to predict 28 day mortality were evaluated. *Results*: The overall mortality was 49.25%. Mechanical ventilation was associated with a higher mortality rate. The levels of IL-6 were significantly higher in the non-survival group and had higher AUC values compared to the other biomarkers. The GCS, SOFA, APACHEII, and SAPS II/III showed superior predictive ability. Combining IL-6 with suPAR, AZU1, and clinical scores SOFA, APACHE II, and SAPS II enhanced prediction accuracy compared with individual biomarkers. *Conclusion*: In our study, IL-6 and SAPS II/III were the most accurate predictors of 28 day mortality for sepsis patients in the ED.

## 1. Introduction

Sepsis represents a significant worldwide health concern accompanied by high rates of mortality and morbidity, requiring costly intensive care (17.158 €–53.951 €/stay) [1,2,3]. Its complex nature and the body’s dysregulated response to infection underline the need for early and accurate diagnostic and prognostic tools in the Emergency Department (ED) [4,5,6,7]. The rapid deterioration of patients with sepsis emphasizes the importance of early detection, continuous monitoring, and precise prognosis to improve outcomes [8]. Despite advancements, an optimal biomarker, or a set of reliable markers for sepsis, remains elusive [9]. Novel biomarkers with high sensitivity, specificity, and bedside monitoring capabilities are essential for enhancing early diagnosis, enabling timely intervention and potentially improving patient outcomes [9,10].

The following clinical indicators have been proposed in the ED to assess the prognosis of patients diagnosed with sepsis: the Quick Sequential Organ Failure Assessment (qSOFA) [5,11,12], Systemic Inflammatory Response Syndrome (SIRS) [4,11], the Sequential Organ Failure Assessment (SOFA) [13,14,15,16], the Acute Physiology and Chronic Health Evaluation II (APACHE II) [17,18], the Simplified Acute Physiology Score II and III (SAPS II/III) [19,20,21], the National Early Warning Score (NEWS) [11], Mortality in Emergency Department Sepsis (MEDS) [21], the Charlson Comorbidity Index (CCI) [18], and the Glasgow Coma Scale (GCS) [22]. Each of these systems integrates clinical and laboratory variables to predict outcomes, yet their effectiveness varies, highlighting the need for additional markers to refine mortality prediction.

Acute phase proteins, cytokines/chemokines, receptor markers, and indicators of endothelial damage, among others, have been discovered as potential diagnostic and prognostic biomarkers in sepsis in recent years. Such biomarkers play pivotal roles in early detection, differential diagnosis, risk assessment, therapy monitoring, and prognosis evaluation of sepsis, including 28 day mortality [23]: procalcitonin (PCT) [24,25], soluble Triggering Receptor Expressed on Myeloid Cells-1 (sTREM-1) [8,9], the soluble form of the urokinase plasminogen activator receptor (suPAR) [9,26], highly sensitive C-reactive protein (hsCRP) [27,28], interleukin-6 (IL-6) [24,29,30], and azurocidin 1 (AZU1) [23,31]. 

PCT, a traditional biomarker, predominantly generated by thyroid C cells under normal conditions, in reaction to infection and inflammation, is notably elevated in the bloodstream, bypassing its usual conversion to calcitonin [25,32]. sTREM-1, primarily expressed on monocytes and neutrophils, enhances the immune response to sepsis by activating Toll-like receptors and by promoting the release of pro-inflammatory cytokines. It was reported to be a critical biomarker for infectious diseases [33,34]. suPAR, expressed on various immune and endothelial cells, indicating immune activation, is linked to an elevated mortality risk, especially in high-risk patients [9,26,35]. Increased levels of acute-phase protein hs-PCR, which is produced in response to IL-6, are correlated with cardiovascular diseases, stroke, all-cause mortality, and potentially future septic events [27,28,36]. Interleukins, specifically IL-6, play a significant role in enhancing immune cell sensitivity, activating cytotoxic responses, and in modulating adaptive immunity. Despite several reports on its use for sepsis prognosis, IL-6 is still incompletely evaluated as a prognosis marker in sepsis and inflammation [29,37,38]. AZU1, produced by polymorphonuclear neutrophils, acts as a chemoattractant and promoter of vascular leakage, with increased plasma levels in septic patients indicating severe disease and potential organ dysfunction [23,31,39].

These biomarkers are providing information on the severity, prognosis, and course of the disease by reflecting various aspects of the immune response, such as inflammation, infection severity, and immune system activation [40].

Integration of these biomarkers with clinical scoring systems may be a way to improve sepsis diagnosis and prognosis in emergency settings, which is important for the timely and effective care of patients with sepsis. This study aimed to evaluate whether combining these biomarkers with clinical scoring systems could enhance the prediction of 28 day mortality in sepsis patients from the ED.

## 2. Materials and Methods

### 2.1. Study Design 

From November 2020 to December 2022, this single-center, observational, prospective study enrolled a consecutive cohort of adult patients with sepsis and septic shock, who were hospitalized in Cluj County Hospital’s Emergency Department, Romania.

The study followed the ethical principles and clinical practice standards outlined in the Helsinki Declaration Ethical Guidelines and EU legislation. The study was approved by the Ethics Committee of the “Iuliu Hațieganu” University of Medicine and Pharmacy Cluj-Napoca (No.139/30.03.2020), and of the ECs of participating hospitals: Cluj Emergency County Hospital (No. 5416/10/25.02.2020), Clinical Hospital for Infectious Diseases Cluj-Napoca (No. 6010/14.04.2021), and Clinical Institute of Urology and Renal Transplantation Cluj-Napoca (No. 03/02.02.2021). Within the first hour after arriving at the emergency department, an informed consent form was signed by the patients or their next of kin.

### 2.2. Study Population Inclusion Criteria

Adult patients aged 18 to 90 years, who met the Sepsis-3 definitions [7] for septic shock and sepsis within one hour of their admission to the ED, were enrolled in the study. According to the current guidelines, sepsis is defined as “a life-threatening organ dysfunction, identified as an acute change in total SOFA score ≥ 2 points, caused by a dys-regulated host response to infection. Septic shock was identified in patients with sepsis with persisting hypotension requiring vasopressors to maintain MAP ≥ 65 mm Hg and having a serum lactate level > 2 mmol/L (18 mg/dL) despite adequate volume resuscitation” [4,7,41].

The following patients were excluded: patients aged under 18 or above 90; pregnant women, patients in custody or deprived of their liberty; patients with associated pathologies such as severe trauma or other types of shocks, acute stroke, burns, pancreatitis, myocardial infarction, pulmonary edema, status asthmaticus, convulsions, poisoning, or drug overdose; patients requiring emergency surgery; and patients in the final stage of cancer. Patients were admitted to the hospital and treated according to standard practices, which included administering broad-spectrum antibiotics, crystalloids, and vasopressors, in line with the Surviving Sepsis Campaign recommendations [5].

Out of 73,939 patients who were referred to the ED during the recruitment process, 488 were diagnosed with sepsis or septic shock. Of these, 67 patients fulfilled the study’s inclusion requirements and were enrolled. Figure 1 presents the patients’ flow chart and outcome after 28 days. Baseline characteristics of the enrolled patients are provided in Table 1.

After admission and initial patient assessment, which included clinical evaluation and laboratory tests (comprising cultures and imaging), severity scores were calculated and a panel of six biomarkers (sTREM-1, hsPCR, PCT, AZU1, suPAR, and IL-6) was analyzed. Patients were monitored for 28 days after ED admission. If discharged from the hospital before, 28 day mortality was determined by a telephone follow-up.

### 2.3. Data Collection

The following information was registered: demographic data, vital signs, symptoms, patient history (comorbidities and recent hospitalization), information on mechanical and non-invasive ventilation, use of vasopressors, standard blood test results, and 28 day mortality. Ten scores (GCS, qSOFA, SIRS, NEWS, CCI, MEDS, SOFA, APACHE II, SAPS II, and SAPS III) were determined using the most severe parameters recorded within the first 3 h in the ED.

### 2.4. Sample Collection and Biomarker Assays

Blood samples were drawn within the first hour after admission from a peripheral vein in a 5 mL serum separator tube with a clot activator, prior to medication or fluid administration. Samples were processed by the hospital laboratory. Blood samples underwent centrifugation at 1000× *g* for 15 min after being drawn, and the supernatant was collected for the assay. After being stored at −20 °C for up to one month, the samples were transferred to a −80 °C freezer until the biomarkers assay. The serum concentrations of the biomarkers were measured using the Sandwich-ELISA immunoassay technique, following the manufacturer’s instructions. Biomarkers were assessed by using ELISA kits: human suPAR (Cat. No: E-EL-H2584, BioVendor, Brno, Czech Republic), human hs-PCR (Cat. No: E-EL-H5134, BioVendor, Brno, Czech Republic), human PCT (Cat. No: E-EL-H1492, BioVendor, Brno, Czech Republic), human sTREM-1 (Cat. No: E-EL-H1596, BioVendor, Brno, Czech Republic), human AZU1 (Cat. No: E-EL-H0540, BioVendor, Brno, Czech Republic), and human IL-6 (Cat. No: E-EL-H0102, BioVendor, Brno, Czech Republic). For the analysis, an Autoanalyzer ELISA Personal Lab ADALTIS (Manufacturer: ADALTIS, Rome, Italy), an ELISA Spectrophotometer LabSystems Multiskan Plus (Manufacturer: LabSystems, Helsinki, Finland), and a Heidolph Shaker Titramax 100 (Manufacturer: Heidolph Instruments GmbH & Co. KG, Schwabach, Germany) were used.

### 2.5. Statistical Analysis

MedCalc^®^ Statistical Software version 22.021 (MedCalc Software Ltd., Ostend, Belgium; https://www.medcalc.org; (accessed on 26 July 2024)) was used to conduct the statistical analysis. The sample size was determined based on an initial small study (n = 12 patients in each group). IL-6 mean values were 470.8 pg/mL in the survival group and 744.7 pg/mL in the non-survival group. For a Type I error (α) of 0.05 and a Type II error (β) of 0.1, a sample size of 27 patients per group was calculated. Frequencies and percentages were used to represent qualitative data. The interquartile range and median were used to characterize quantitative data (non-normal distribution). Whenever appropriate, the chi-square test or the Mann–Whitney U test was used to compare the groups. To evaluate the precision of serum biomarkers and clinical severity scores in predicting mortality within 28 days, Receiver Operating Characteristic (ROC) curve analysis was conducted. Statistical significance was attained when the *p* value was <0.05.

## 3. Results

In the study group, the overall mortality rate was 49.25%. The requirement for mechanical ventilation and elevated lactate levels (*p* < 0.001) were identified as statistically significant factors associated with a higher mortality rate. Twelve patients (36.4%) from the non-survival group presented to the ED with septic shock. Out of the 67 admitted patients, five presented with COVID-19 and one did not survive. No significant differences were observed between the two groups in terms of comorbidities or symptoms (Table 1). Cardiovascular disease was the most common comorbidity, prevalent in 80% of the patients. Primary sites of infection leading to sepsis among these patients were urinary, skin, soft tissue, and respiratory. Patients in the ED received either one or two antibiotics in accordance with the hospital’s guidelines. Most recommended combinations of antibiotics included beta-lactams and vancomycin (Table 1). We adhered to the Surviving Sepsis Campaign guidelines [5] in 85% of the cases. Reasons for not adhering to guidelines in all cases included very crowded periods in the ED, when some patients were either handed over to the ICU or admitted to regular wards before full implementation of the guideline could be completed. Antibiotics were administered in these wards soon after patient admission.

Table 2 displays the median values of sepsis scores and plasma concentrations of selected biomarkers in study groups at admission to the ED.

As it can be seen in Table 2, only the serum levels of IL-6 differed significantly between the two groups, with the survival group showing lower values (290.4 pg/mL, *p* < 0.001). In terms of predicting 28 day mortality, the non-survival group had significantly higher scores for the qSOFA, SAPS II and III, SOFA, and APACHE II.

The ROC curve analysis of selected biomarkers and scores in predicting 28 day mortality are presented in Table 3.

Table 3 shows that all combinations, including IL-6, were significantly different between survivors and non-survivors. Combining suPAR with IL-6 did not significantly modify prognostic accuracy compared to the evaluation of IL-6 alone (AUC = 0.74, *p* < 0.001 vs. AUC = 0.73, *p* < 0.001). However, the simultaneous assessment of IL-6 and AZU1 demonstrated superiority in predicting 28 day mortality compared to using IL-6 alone (AUC = 0.78, *p* < 0.001 vs. AUC = 0.73, *p* < 0.001). Both SAPS II and SAPS III scores showed increased AUC values and high sensitivity in predicting 28 day mortality. However, the addition of other scoring systems or biomarkers to SAPS II did not improve the predictive ability for 28 day mortality compared to using SAPS II alone or the combination of IL-6 and AZU1.

Finally, the AUCs for IL-6 vs. SOFA + SAPS II + APACHE II and IL-6 + SOFA + SAPS II + APACHE II vs. SOFA + SAPS II + APACHE II were compared. The AUCs did not differ significantly (*p* = 0.173 and *p* = 0.461, respectively).

## 4. Discussion

Our study investigated the potential of various acute phase proteins, chemokines, receptor markers, and indicators of endothelial damage, along with severity scoring systems and their combinations, to predict 28 day mortality in patients diagnosed with septic shock and sepsis at admission to the ED. Studies [2,4,5,42,43,44] have reported a high mortality rate among sepsis patients, ranging from 20 to 60% despite significant progress in medical science; in our study, the overall mortality was 49.9%. Upon arrival at the Emergency Department, 36.4% of non-survivors were identified with septic shock, a number that increased to 78.8% in the following hours. The median age was 78. Multiple comorbidities, such as cardiovascular disease, diabetes, or obesity have likely contributed to a high mortality rate. Our data aligns with the existing literature, confirming that increased lactate levels in patients with sepsis and septic shock, resulting from metabolic distress in response to the acute event, are associated with higher mortality rates [45]. Other studies have found that, in addition to increased mortality, elevated lactate levels are also linked to prolonged recovery periods and higher rates of readmission [24,46]. Thus, the focus on early detection and accurate diagnosis is essential, as prompt intervention and appropriate treatment may improve prognosis.

Despite all these advances, there is a critical lack of validated prognostic tools for assessing mortality risk in sepsis patients in the ED. Most studied biomarkers like PCT and hsCRP are not globally endorsed due to inadequate specificity and sensitivity [4,36,47]. Considering the critical role of the ED in the early detection of sepsis and initiation of treatment, it is important to explore new biomarkers and assess clinical scores to improve the prediction of 28 day mortality in septic patients, with the goal of significantly improving patient outcomes and management [13].

IL-6 was the only biomarker in our study that demonstrated statistically notable variations when comparing survivors and non-survivors, with lower levels observed in survivors and a cut-off value of >538 pg/mL. Our results are consistent with other findings [48], while others reported significantly higher threshold IL-6 levels up to 1600 pg/mL compared with ours [49]. IL-6 is rapidly expressed (often within 2 h) by various cell types in response to infections and tissue injuries, playing a significant part in the sepsis-related systemic response to infection [36]. Higher IL-6 levels were associated with the extent of organ dysfunction and influenced mortality rates [25,29,30,37]. Despite varying thresholds, IL-6 levels are considered a reliable predictive biomarker for the determination of a 28 day mortality prognosis [24,29,37,38]. A ROC curve analysis of our data also indicated that IL-6 is a standalone predictor of 28 day mortality in our study, consistent with findings from other studies [24,48]. It is worth noting that in another emergency setting, IL-6 exhibited slightly reduced performance, with a threshold value of 2580.50 pg/mL [48]. Despite this variability, these findings indicate that IL-6 may be used as a potential biomarker to predict 28-day mortality in sepsis patients [24,25,30]. The lower IL-6 levels in our study may be due to various factors, including the early or late stages of sepsis on admission, when IL-6 peaks can be missed due to their transient nature. Additionally, patient demographics like age, comorbidities, and genetics, as well as pre-hospital treatments with fluids or steroids, could also affect the production of IL-6 [37].

In our study, the levels of suPAR, AZU1, PCT, hsCRP, and sTREM-1 did not vary significantly between the two groups. Consequently, an Area Under the Curve (AUC) analysis does not confirm a significant value in predictions for 28 day mortality. In this study, the biomarker samples from patients with sepsis were collected within the first hour of their arrival at the ED. It is important to mention that during this timeframe, it is possible that some patients had sepsis in its early stages, when serum concentrations had not yet begun to increase. Additionally, the half-life of biomarkers can vary significantly, potentially causing us to miss their peak levels and thus underestimating their prognostic value [50].

In a prospective study on 120 patients with bacterial bloodstream infections, the AUC values were 0.94 for PCT, 0.78 for hsCRP, and 0.99 for their combination, showing a significant diagnostic value in the short-term prognosis prediction [51]. In a meta-analysis focused on mortality prediction among sepsis patients, the AUC for suPAR ranged from 0.67 to 0.78, with higher concentrations in non-survivors. Similarly, the AUC for sTREM-1 ranged between 0.44 and 0.85, similarly with elevated levels in non-survivors [52].

The lack of a predictive value for mortality of procalcitonin in our study is similar with findings from other studies [53,54]. However, PCT continues to be a key marker in managing septic patients, particularly in guiding the cessation of antibiotic therapy [4,55,56].

The combination of IL-6 with AZU1 resulted in an increased sensitivity, reaching 90.91%. However, this combination did not significantly improve the prognostic value, achieving an AUC of 0.78, which is marginally higher than the 0.73 AUC for IL-6 alone. 

In our study, AZU1 was found to have a limited prognostic value in septic patients, which contrasts with findings from other studies reporting notably higher plasma AZU1 levels in sepsis associated with organ dysfunction. This suggests its potential utility as a promising biomarker for diagnosis and prediction in severe cases of sepsis [23,31]. 

Since each individual biomarker has a limited predictive significance, the next step was to evaluate whether combining them with clinical scoring systems would result in a better tool for predicting mortality. Thus, a combination of IL-6, suPAR and SOFA, SAPS II, and APACHE II showed a higher AUC value compared to IL-6 alone or conventional scoring systems alone.

In the Intensive Care Unit (ICU), SOFA, APACHE II, SAPS II and III are the most utilized scoring systems. Our research, consistent with findings from other studies, indicates that SAPS II and III outperformed APACHE II and SOFA in predicting 28 day mortality [19,20].

However, combining SAPS II with other scoring systems or biomarkers did not improve the predictive accuracy for 28 day mortality compared to using SAPS II alone or the combination of IL-6 and AZU1. This suggests that SAPS II remains a reliable predictor of 28 day mortality [19,20].

To our knowledge, only a limited number of studies focused on the simultaneous analysis of IL-6, AZU1, sTREM-1, suPAR, hsPCR, and PCT. IL-6 demonstrated a predictive value close to that of complex scoring systems like SOFA, APACHE II, and SAPS II/III, suggesting its potential as a simpler yet effective biomarker for disease prognosis [29,30,31,32,33,34,35,36,37]. Variability in patient population, including differences in the etiology of sepsis, underlying health conditions, age, genetic factors, or the presence of comorbidities, could affect biomarker levels and their prognostic value.

Our study has several limitations. First, it is a monocentric study, which may limit the generalizability of our findings to other settings and populations. The sample size was relatively small, a constraint exacerbated by the COVID-19 pandemic restrictions during the study period. These restrictions limited access to hospitals and wards, thereby restricting our ability to recruit a larger number of patients and to obtain comprehensive data. Additionally, follow-ups were hindered by the pandemic, as patients and their relatives were often reticent to participate in studies due to concerns related to the virus. Also, the small sample size may not have been sufficient to capture the full variability and range of biomarkers that larger studies have observed. Another limitation may be the early assessment of biomarkers (soon after the admission to the ED) that may have influenced plasma levels of certain biomarkers where an increase may have been registered later in time. These limitations may have impacted the robustness and generalizability of our results.

## 5. Conclusions

Our study demonstrated that IL-6 could be a valuable biomarker for predicting 28 day mortality in sepsis compared with the other biomarkers. Of the common predictive scores, the strongest performance in predicting 28 day mortality was achieved by SAPS II and SAPS III. Integrating IL-6 with these clinical scores may enhance the prognostic accuracy for sepsis and septic shock patients at admission to the ED. More large multicenter trials are necessary to validate the current findings and assess their impact on patient management and outcomes.

## Figures and Tables

**Figure 1 medicina-60-01333-f001:**
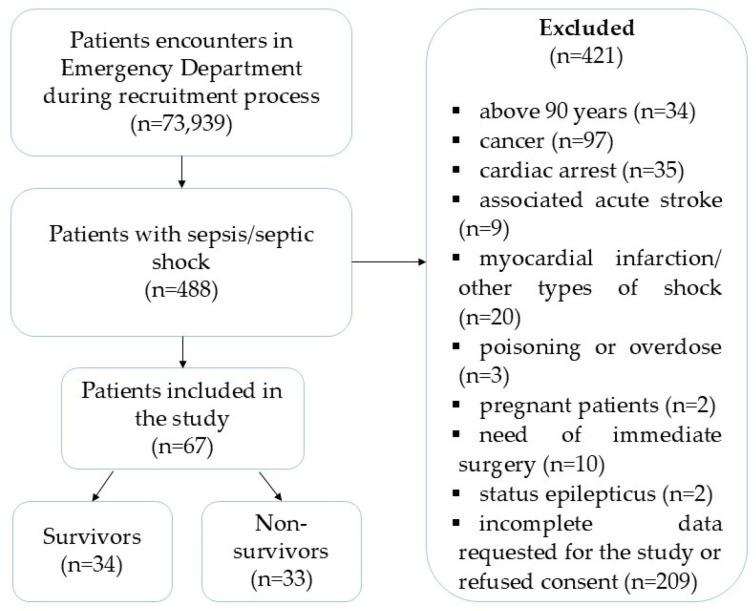
Flowchart of the patients included in the study.

**Table 1 medicina-60-01333-t001:** Demographic characteristics of the study groups on admission to ED.

Characteristics [Median (IQR)]		Survivors (Group = 34)	Non-Survivors (Group = 33)	*p*
Gender, n (%)				
Male		16 (47.1)	12 (36.4)	0.52 *
Female		18 (52.9)	21 (63.6)	
Age, years		72 (63.75–81.25)	78 (65.5–85)	0.11 **
Recent hospitalization, n (%)		6 (17.6)	7 (21.2)	0.95 *
ICU stay (days)		2 (1–11.75)	5 (2–9)	0.48 **
Oxygen mask, n (%)		18 (52.9)	17 (51.5)	1.00 *
Mechanical ventilation, n (%)		3 (8.8)	17 (51.5)	**<0.001** *
Non-invasive ventilation, n (%)		0 (0)	2 (6.1)	0.23 *
Vasoactive, n (%)		7 (20.6)	12 (36.4)	0.24 *
Antibiotics administered, (n%)	1	22 (75.9)	24 (85.7)	0.54 *
	2	7 (24.1)	4 (14.3)	
**Parameters**				
Heart rate		110 (93.5–115.25)	115.25 (100.5–124.5)	0.14 **
Mean arterial pressure (mmHg)		63 (56.75–72.25)	60 (53–65.5)	0.14 **
Temperature (˚C)		38 (36–39)	37 (36–38)	0.01 **
Oxygen Saturation (%)		94.5 (88.75–97)	94 (86–97)	0.50 **
Lactate (mmol/L)		2 (1–3)	4 (2–7.5)	**<0.001** **
**Comorbidities, n (%)**				
Cardiovascular disease		27 (79.4)	29 (87.9)	0.54 *
Diabetes		19 (55.9)	17 (51.5)	0.91 *
Chronic kidney disease		9 (26.5)	7 (21.2)	0.82 *
Chronic lung disease		9 (26.5)	5 (15.2)	0.40 *
Obesity		15 (44.1)	11 (33.3)	0.51 *
Neuropsychiatry		14 (41.2)	21 (63.6)	0.11 *
**Site of sepsis, n (%)**				
Neurologic		0 (0)	1 (3)	0.49 *
Respiratory		15 (44.1)	21 (63.6)	0.17 *
Cardiac		0 (0)	1 (3)	0.49 *
Digestive		8 (23.5)	4 (12.1)	0.36 *
Skin and soft tissue		13 (38.2)	13 (39.4)	1.00 *
Urinary		24 (70.6)	20 (60.6)	0.54 *

ICU—Intensive Care Unit; * chi-square test; ** the Mann–Whitney U Test.

**Table 2 medicina-60-01333-t002:** Blood biomarkers and severity scores in the study groups.

Parameters †		Survival Group	Non-Survival Group	*p*
**Biomarkers**				
**sTREM-1 (pg/mL)**		185.3 (80.70–722.90)	274.5 (94.55–967.95)	0.59 **
**hsCRP (pg/mL)**		26.05 (16–29.95)	18.4 (15.75–24.30)	0.10 **
**PCT (pg/mL)**		9.4 (2.70–20.67)	15.2 (4.70–56.10)	0.09 **
**AZU1 (ng/mL)**		8.3 (7.47–9.12)	7.6 (6.90–8.55)	0.05 **
**suPAR (ng/mL)**		7421 (6060.75–8900.25)	8256 (6995–9601)	0.05 **
**IL-6 (pg/mL)**		290.4 (76.30–529.22)	694 (346.50–858.30)	**<0.001** **
**Scores, n (%)**				
**SIRS**	2	7 (20.60)	6 (18.20)	0.52 *
	3	14 (41.20)	18 (54.50)	
	4	13 (38.20)	9 (27.20)	
**qSOFA**	1	8 (23.50)	2(6.10)	**0.001** *
	2	18 (52.90)	8 (24.20)	
	3	8 (23.50)	23 (69.70)	
**GCS**		15 (12.75–15)	11 (6.50–14.50)	**<0.001** **
**CCI**	CCI Score	6.5 (4–9)	7 (6–9)	0.34 **
	CCI %	1 (0–53)	0 (0–2)	0.14 **
**NEWS**		10 (7–12)	13 (10.5–15)	0.003 **
**MEDS score**		12.5 (11–16)	16 (11–17)	0.14 **
**SOFA**		5 (3–9.25)	10 (7–13)	**0.001** **
**APACHE II**	APACHEII score	21 (14.75–24.25)	26 (21.50–33)	**<0.001** **
	APACHE II %	40 (22.50–46.75)	55 (40–73)	**<0.001** **
**SAPS II**	SAPS II score	46.5 (39.75–56.75)	66 (56–85)	**<0.001** **
	SAPS II %	38.1 (27.12–61.37)	78.5 (59.8–95)	**<0.001** **
**SAPS III**	SAPS III score	64.5 (59–71.25)	81 (69.50–95)	**<0.001** **
	SAPS III %	45 (33.87–58.97)	75 (56–88.55)	**<0.001** **

† [median (IQR)], sTREM-1—soluble Triggering Receptor Expressed on Myeloid Cells-1, hsCRP—high-sensitivity C-reactive protein, PCT—procalcitonin, AZU1—azurocidin 1, suPAR—soluble urokinase plasminogen activator, IL-6—interleukin-6, SIRS—Systemic Inflammatory Response Syndrome, qSOFA—Quick Sequential Organ Failure Assessment, GCS—Glasgow Coma Scale, CCI—Charlson Comorbidity Index, NEWS—National Early Warning Score, MEDS—Mortality in Emergency Department Sepsis, SOFA—Sequential Organ Failure Assessment, APACHE II—Acute Physiology and Chronic Health Evaluation II, SAPS II and III—Simplified Acute Physiology Score II and III; * chi-square test; ** the Mann–Whitney U Test.

**Table 3 medicina-60-01333-t003:** Prognostic value of biomarkers, clinical scores, and combinations of clinical scores with biomarkers in predicting 28 day mortality on admission to ED.

	AUC (95% CI)	Cutoff	Se (95% CI)	Sp (95% CI)	*p*
**Biomarkers**					
IL-6 (pg/mL)	0.73 (0.61–0.83)	>538	63.64 (45.1–79.6)	79.41 (62.1–91.3)	**<0.001**
suPAR (ng/mL)	0.63 (0.51–0.75)	>7447	69.70 (51.3–84.4)	58.82 (40.7–75.4)	0.04
PCT (pg/mL)	0.62 (0.49–0.73)	>19.8	48.48 (30.8–66.5)	76.47 (58.8–89.3)	0.08
hsCRP (pg/mL)	0.61 (0.48–0.73)	≤24.9	81.82 (64.5–93.0)	52.94 (35.1–70.2)	0.10
sTREM-1 (pg/mL)	0.53 (0.41–0.66)	>189	63.64 (45.1–79.6)	52.94 (35.1–70.2)	0.59
AZU1 (ng/mL)	0.63 (0.5–0.7)	≤7.7	63.64 (45.1–79.6)	67.65 (49.5–82.6)	0.05
**Scores**					
GCS	0.75 (0.63–0.85)	≤12	66.67 (48.2–82.0)	76.47 (58.8–89.3)	**<0.001**
NEWS	0.71 (0.58–0.81)	>11	69.7 (51.3–84.4)	73.53 (55.6–87.1)	**0.001**
SOFA	0.74 (0.62–0.84)	>6	81.82 (64.5–93)	58.82 (40.7–75.4)	**<0.001**
APACHE II	0.77 (0.66–0.87)	>23	69.7 (51.3–84.4)	73.53 (55.6–87.1)	**<0.001**
SAPS II	0.82 (0.70–0.90)	>43.8	93.94 (79.8–99.3)	61.76 (43.6–77.8)	**<0.001**
SAPS III	0.82 (0.71–0.90)	>67	90.91 (75.7–98.1)	67.65 (49.5–82.6)	**<0.001**
**Combinations**					
*IL-6 + suPAR*	0.74 (0.62–0.84)	>0.54	66.67 (48.2–82)	76.47 (58.8–89.3)	**<0.001**
*IL-6 + AZU*	0.78 (0.66–0.87)	>0.34	90.91 (75.7–98.1)	52.94 (35.1–70.2)	**<0.001**
*IL-6 + SOFA + SAPSII + APACHE II*	0.85 (0.75–0.93)	>0.61	63.64 (45.1–79.6)	94.12 (80.3–99.3)	**<0.001**
*IL-6 + AZU + SOFA + SAPSII + APACHE II*	0.82 (0.76–0.94)	>0.53	75.76 (57.7–88.9)	82.35 (65.5–93.2)	**<0.001**
*IL-6 + suPAR + AZU + SOFA + SAPSII+ APACHE II*	0.86 (0.76–0.93)	>0.53	75.76 (57.7–88.9)	82.35 (65.5–93.2)	**<0.001**
*SOFA + SAPSII + APACHEII*	0.83 (0.72–0.91)	>0.4	81.82 (54.5–93)	70.59 (52.5–84.9)	**<0.001**

AUC—area under the curve, CI—confidence interval, Se—sensitivity, Sp—specificity, sTREM-1—soluble Triggering Receptor Expressed on Myeloid Cells-1, hsCRP—high-sensitivity C-reactive protein, PCT—procalcitonin, AZU1—azurocidin 1, suPAR—soluble urokinase plasminogen activator, IL-6—interleukin-6, GCS—Glasgow Coma Scale, NEWS—National Early Warning Score, SOFA—Sequential Organ Failure Assessment, APACHE II—Acute Physiology and Chronic Health Evaluation II, SAPS II and III—Simplified Acute Physiology Score II and III; ROC analysis.

## Data Availability

Upon reasonable request, the datasets from the current study will be made available by the corresponding author.

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
