# Peer review of "Can We Improve Mortality Prediction in Patients with Sepsis in the Emergency Department?"

_medicina, 2024, doi:10.3390/medicina60081333_

Round 1

Reviewer 1 Report

Comments and Suggestions for Authors

What are Sepsis-3 criteria ? 

What saturation did the patients have? Did some of them also receive oxygen on a face mask? (not just mechanical/noninvasive ventilation?) 

Why the lactate level was not measured?

What was the outcome of the patients?

What broad-spectrum antibiotics were used? Did all study participants received antibiotics?

Was the 'Sepsis Six' guideline pathway respected? 

How many days did the patients stay in the ICU? 

Another limitation would be that the study is monocentric, please discuss this aspect. 

To improve the manuscript, please adress other relevant studies from literature  regarding biomarkers in septic shock:

a)    https://doi.org/10.3390/clinpract14030065

b)    https://doi.org/10.3390/biomedicines8110494

c)     https://doi.org/10.3390/biomedicines11082149

d)    https://doi.org/10.3390/app12031419

e)    https://doi.org/10.2147/IJGM.S464892

Comments on the Quality of English Language

Minor editing of English language required

Author Response

Thank you very much for taking the time to review our manuscript!

  1. What are Sepsis-3 criteria?

Response 1: Thank you for this comment. We have rephrased the relevant paragraph by including the current Sepsis-3 definitions, as follows:

Adult patients aged 18 to 90 years, who met the Sepsis-3 definitions [7] for septic shock and sepsis within one hour of their admission to the ED were enrolled in the study. According to the current guidelines, sepsis is defined as” a life-threatening organ dysfunction, identified as an acute change in total SOFA score ≥2 points, caused by a dysregulated host response to infection. Septic shock was identified in patients with sepsis with persisting hypotension requiring vasopressors to maintain MAP ≥65 mm Hg and having a serum lactate level >2 mmol/L (18 mg/dL) despite adequate volume resuscitation”. [4, 7, 39]

Rows: 114 – 119, page 3, paragraph 3

  1. What saturation did the patients have? Did some of them also receive oxygen on a face mask? (not just mechanical/noninvasive ventilation?)

Response 2: Thank you for this well-deserved observation. We have added the saturation data and information about oxygen administration via face mask to Table 1 (page 5).

  1. Why the lactate level was not measured?

Response 3: Thank you for your observation. Lactate levels were measured, and we have now included this data in Table 1, underlining the updated information. We also provided a detailed interpretation of its significance within the study in the Results and Discussion section.

Rows: 177 – 179, page 4, paragraph 4 and row 228 – 232, page 8, paragraph 3

  1. What was the outcome of the patients?

Response 4: We assessed the 28-day mortality of the patients and divided the study group into survivors and non-survivors based on this outcome. We also assessed and compared ICU LOS between these study groups and data were provided in Table 1 (page 5).

  1. What broad-spectrum antibiotics were used? Did all study participants received antibiotics?

Response 5: Thank you for your question regarding the use of broad-spectrum antibiotics in our study. Antibiotherapy was administered according to hospital guidelines, considering factors such as the site of sepsis, recent hospitalizations, comorbidities, and the clinical judgment of the attending physician. However, it is important to note that this was not the primary focus of our study, as blood samples were collected prior to the administration of antibiotics.

Data on antibiotics were added in Table 1 (page 5). We also specified which was the most used combination of antibiotics in Results section. Reasons for not administering antibiotics in all patients have been provided in Results section.

Rows: 186- 187, page 4, paragraph 4

  1. Was the 'Sepsis Six' guideline pathway respected?

Response 6: Thank you for your question regarding adherence to the 'Sepsis Six' guideline pathway. We adhered to the 'Sepsis Six / 2016 Surviving Sepsis Campaign guidelines (considering that we developed the study protocol in 2020) in 85% of the cases. Reasons for not adhering to guidelines in all cases included very crowded periods in ED, when some patients were either handed over to the ICU or admitted to regular wards (where the case) before full implementation of the guideline could be completed. Antibiotics were administered of these wards soon after patient admission. This paragraph was added in the manuscript as well in Results section.

Rows: 187 – 192, page 4

  1. How many days did the patients stay in the ICU?

Response 7: We have added the data regarding ICU LOS in Table 1 (page 5).

  1. Another limitation would be that the study is monocentric, please discuss this aspect.

Response 8: Thank you for this well-deserved suggestion. We have rephrased and elaborated on the limitation section. Bellow is the revised limitation section:

” Our study has several limitations. First, it is a monocentric study, which may limit the generalizability of our findings to other settings and populations. The sample size was relatively small, a constraint exacerbated by the COVID-19 pandemic restrictions during the study period. These restrictions limited access to hospitals and wards, thereby restricting our ability to recruit a larger number of patients and to obtain comprehensive data. Additionally, follow-up was hindered by the pandemic, as patients and their relatives were often reticent to participate in studies due to concerns related to the virus. Also, the small sample size may not have been sufficient to capture the full variability and range of biomarkers that larger studies have observed. Another limitation may be the early assessment of biomarkers (soon after the admission to ED) that may have influenced plasma levels of certain biomarkers where an increase may have been registered later in time. These limitations may have impacted the robustness and generalizability of our results.”

Rows: 306 – 317, page 10, paragraph 4

  1. To improve the manuscript, please address other relevant studies from literature regarding biomarkers in septic shock:

Response 9: Thank you for your suggestions. We have reviewed and added relevant and updated literature on biomarkers and scores in septic shock.

Rows: 352 - 502, page 11 – 14

  1. Minor editing of English language required

Response 10: We have revised the English language in the manuscript and hopefully this aspect is improved. Additionally, several other minor English language corrections have been made throughout the manuscript to enhance clarity.

Reviewer 2 Report

Comments and Suggestions for Authors

Thank you for the opportunity to read an interesting manuscript. The topic raised is very important due to the risks resulting from sepsis and limited diagnostics in ED conditions. I ask the authors to take into account the following comments:

1) Figure 1 has incorrect calculations (e.g. 430 out of 488 cases were excluded, which should have given a result of 58, not 67).

2) Could the period of the study have affected the biomarker results? Many patients had COVID-19 at the time. Due to the time of the pandemic, this aspect should be addressed in the work.

3) The P value should be unified in all tables and descriptions (e.g. to 2 decimal places). I suggest adding the statistical test used under the tables.

4) The limitations of the study should be much more extensive. The main confounding factors should be listed.

5) The references contain many publications that cannot be considered current (e.g. from 2000 and 2001). I suggest supplementing the references with several current studies that directly refer to the authors' study:

a) Mourya V, Gupta R, Yadav A, Yadav R. Lactate / albumin ratio as a prognostic tool for risk stratification in septic patients admitted to ICU. Crit. Care Innov. 2023; 6(4): 11-22. DOI: 10.32114/CCI.2023.6.4.11.22

b) Kumar DS, Wasnik SB, Yadav A, Yadav R. Association of glycosylated hemoglobin with mortality of patients in intensive care unit: a prospective observation study. Crit. Care Innov. 2024; 7(1): 24-33. DOI: 10.32114/CCI.2024.7.1.24.33

Author Response

Thank you very much for taking the time to review our manuscript!

  • Figure 1 has incorrect calculations (e.g. 430 out of 488 cases were excluded, which should have given a result of 58, not 67).

Response 1: Thank you for your observation regarding Figure 1. The discrepancy was due to the initial inclusion of patients with more than one exclusion criterion, which was not clearly explained. We have updated the table to now reflect only one exclusion criterion for each excluded patient, ensuring clarity and accuracy. Thank you for bringing this to our attention.

Page 3 – Figure 1

  • Could the period of the study have affected the biomarker results? Many patients had COVID-19 at the time. Due to the time of the pandemic, this aspect should be addressed in the work.

Response 2: Thank you for this comment regarding the potential impact of the study period on the biomarker results, particularly considering the prevalence of COVID-19 at the time. We had a small number of patients admitted with COVID-19 (5 patients in total), with one non-surviving patient among them. Given this limited number, we did not consider the presence of COVID-19 to have a significant impact on the overall study results. Data on patients having COVID-19 in our study have been provided in Results section.

    Rows: 180 – 181, page 4, paragraph 4

  • The P value should be unified in all tables and descriptions (e.g. to 2 decimal places). I suggest adding the statistical test used under the tables.

Response 3: We have reviewed and corrected the data as suggested. Additionally, we added the statistical tests used under each table for clarity.

Table 1 - Page 5

Table 2 - Page 6

Table 3 - Page 7

  • The limitations of the study should be much more extensive. The main confounding factors should be listed.

Response 4: We appreciate your suggestion. We have revised the limitations section to be more comprehensive. We have outlined the primary issues and confounding factors that could impact the validity and generalizability of our findings. Below is a summary of the key changes we made: 

  • Monocentric study
  • Relatively small sample size
  • COVID 19 pandemic restrictions
  • Early assessment of the biomarkers

Row: 306 – 317, page 10, paragraph 4

  • The references contain many publications that cannot be considered current (e.g. from 2000 and 2001). I suggest supplementing the references with several current studies that directly refer to the authors' study:

Response 5: Thank you for your suggestion regarding the references. We have reviewed and revised the bibliography as suggested, supplementing it with several recent studies.

   Rows: 352 - 502, page 11 – 14